# Nonlinear deformation and run-up of single tsunami waves of positive polarity: numerical simulations and analytical predictions

Ahmed A. Abdalazeez [1], Ira Didenkulova [1, 2], Denys Dutykh [3]

[1] Department of Marine Systems, Tallinn University of Technology, Akadeemia tee 15A, Tallinn 12618, Estonia
[2] Nizhny Novgorod State Technical University n.a. R.E. Alekseev, Minin str. 24, Nizhny Novgorod 603950, Russia
[3] Univ. Grenoble Alpes, Univ. Savoie Mont Blanc, CNRS, LAMA, Chambéry, 73000, France

*Correspondence to*: Ahmed A. Abdalazeez (ahabda@ttu.ee)

**Abstract.** The estimate of individual wave run-up is especially important for tsunami warning and risk assessment as it allows to evaluate the inundation area. Here as a model of tsunami we use the long single wave of positive polarity. The period of such wave is rather long which makes it different from the famous Korteweg-de Vries soliton. This wave nonlinearly deforms during its propagation in the ocean, what results in a steep wave front formation. Situations, when waves approach the coast with a steep front are often observed during large tsunamis, e.g. 2004 Indian Ocean and 2011 Tohoku tsunamis. Here we study the nonlinear deformation and run-up of long single waves of positive polarity in the conjoined water basin, which consists of the constant depth section and a plane beach. The work is performed numerically and analytically in the framework of the nonlinear shallow water theory. Analytically, wave propagation along the constant depth section and its run-up on a beach are considered independently without taking into account wave interaction with the toe of the bottom slope. The propagation along the bottom of constant depth is described by Riemann wave, while the wave run-up on a plane beach is calculated using rigorous analytical solutions of the nonlinear shallow water theory following the Carrier-Greenspan approach. Numerically, we use the finite volume method with the second order UNO2 reconstruction in space and the third order Runge-Kutta scheme with locally adaptive time steps. During wave propagation along the constant depth section, the wave becomes asymmetric with a steep wave front. Shown, that the maximum run-up height depends on the front steepness of the incoming wave approaching the toe of the bottom slope. The corresponding formula for maximum run-up height, which takes into account the wave front steepness, is proposed.

## 1. Introduction

Evaluation of wave run-up characteristics is one of the most important tasks in coastal oceanography especially when estimating tsunami hazard. This knowledge is required as for planning coastal structures and protection works, as for short-term tsunami forecast and tsunami warning. Its importance is also confirmed by a number of scientific papers, e.g. see recent works (Tang et al. 2017; Touhami and Khellaf 2017; Zainali et al. 2017; Raz et al. 2018; Yao et al. 2018).

The general solution of the nonlinear shallow water equations on a plane beach was found by Carrier and Greenspan (1958) using the hodograph transformation. Later on many other authors found specific solutions for different types of waves climbing the beach, see, for instance, (Pedersen and Gjevik 1983; Synolakis 1987; Synolakis et al. 1988; Mazova et al. 1991; Pelinovsky and Mazova 1992; Tadepalli and Synolakis 1994; Brocchini and Gentile 2001; Carrier et al. 2003; Kânoglu 2004; Tinti and Tonini 2005; Kânoglu and Synolakis 2006; Madsen and Fuhrman 2008; Didenkulova et al. 2007; Didenkulova 2009; Madsen and Schaffer 2010).

Many of these analytical formulas have been validated experimentally in laboratory tanks (Synolakis 1987, Li and Raichlen 2002; Lin et al. 2009; Didenkulova et al. 2013). For most of them, the solitary waves have been used. The soliton is rather easy to generate in the flume, therefore, laboratory studies of run-up of solitons are the most popular. However, (Madsen et al. 2008) pointed out that the solitons are inappropriate to describe the real tsunami and proposed to use waves of longer duration than solitons and downscaled records of real tsunami. Schimmels et al. (2016) and Sriram et al. (2016) generated such long waves in the Large Wave Flume of Hannover (GWK FZK) using the piston type of wave maker while McGovern et al. (2018) did it using the pneumatic wave generator.

It should be mentioned that the shape of tsunami varies a lot depending on its origin and the propagation path. One of the best examples of tsunami wave shape variability is given in Shuto (1985) for the 1983 Japan Sea tsunami, where the same tsunami event resulted in very different tsunami approaches in different locations along Japanese coast. These wave shapes included: single positive pulses, undergoing both surging and spilling breaking scenarios, breaking bores, periodic wave trains, surging and breaking as well, a sequence of two or three waves and undular bores. This is why there is no such term as "typical tsunami wave shape", and therefore in the papers on wave run-up cited above many different wave shapes, such as single pulses, *N*-waves, periodic symmetric and asymmetric wave trains, are considered. In this paper, we focus on the nonlinear deformation and run-up of long single pulses of positive polarity on a plane beach.

A similar study was performed for periodic sine waves (Didenkulova et al. 2007; Didenkulova 2009). It was shown that the run-up height increases with an increase in the wave asymmetry (wave front steepness) which is a result of nonlinear wave deformation during its propagation in a basin of constant depth. It was found analytically that the run-up height of this nonlinearly deformed sine wave is proportional to the square root of the wave front steepness. Later on, this result was also confirmed experimentally (Didenkulova et al. 2013).

It should be noted that these analytical finding also match tsunami observations. Steep tsunami waves are often witnessed and reported during large tsunami events, such as 2004 Indian Ocean and 2011 Tohoku tsunamis. Sometimes the wave, which approaches the coast, represents a "wall of water" or a bore, which is demonstrated by numerous photos and videos of these events.

The nonlinear steepening of the long single waves of positive polarity has also been observed experimentally in (Sriram et al. 2016), but its effect on wave run-up has not been studied yet. In this paper, we study this effect both analytically and numerically. Analytically, we apply the methodology developed in (Didenkulova 2009; Didenkulova et al. 2014), where we consider the processes of wave propagation in the basin of constant depth and the following wave run-up on a plane beach

independently, not taking into account the point of merging of these two bathymetries. Numerically, we solve the nonlinear shallow water equations.

The paper is organized as follows. In Section 2, we give the main formulas and briefly describe the analytical solution. The numerical model is described and validated in Section 3. The nonlinear deformation and run-up of the long single wave of positive polarity is described in Section 4. The main results are summarized in Section 5.

## 2. Analytical solution

We solve the nonlinear shallow water equations for the bathymetry shown in Fig. 1:

$$\frac{\partial u}{\partial t} + u\frac{\partial u}{\partial x} + g\frac{\partial \eta}{\partial x} = 0 \,, \tag{1}$$

$$\frac{\partial \eta}{\partial t} + \frac{\partial}{\partial x}\left[\left(h(x)+\eta\right)u\right] = 0 \,. \tag{2}$$

Here $\eta(x, t)$ is the vertical displacement of the water surface with respect to the still water level, $u(x,t)$ – depth-averaged water flow, $h(x)$ – unperturbed water depth, $g$ is the gravitational acceleration, $x$ is the coordinate directed onshore, and $t$ is time. The system of Eqs.(1),(2) is solved independently for the two bathymetries shown in Fig. 1: a basin of constant depth $h_0$ and length $X_0$ and a plane beach, where the water depth $h(x) = - x \tan\alpha$.

Eqs. (1),(2) can be solved exactly for a few specific cases. In the case of constant depth, the solution is described by the Riemann wave (Stoker 1957). Its adaptation for the boundary problem can be found in Zahibo et al. (2008). In the case of a plane beach, the corresponding solution was found by Carrier and Greenspan (1958). Both solutions are well-known and widely used and we do not reproduce them here, but just provide some key formulas.

As already mentioned, during its propagation along the basin of constant depth $h_0$, the wave transforms as a Riemann wave (Zahibo et al. 2008):

$$\eta(x,t) = \eta_0\left[t - \frac{x+X_0+L}{V(x,t)}\right], \tag{3}$$

$$V(x,t) = 3\sqrt{g\left[h_0 + \eta(x,t)\right]} - 2\sqrt{gh_0} \,, \tag{4}$$

where $\eta_0(x = -L - X_0, t)$ is the water displacement at the left boundary. After the propagation over the section of constant depth $h_0$, the incident wave has the following shape:

$$\eta_{X0}(t) = \eta_0\left[t - \frac{X_0}{V(x,t)}\right], \qquad V_{X0}(t) = 3\sqrt{g\left[h_0 + \eta_{X0}(t)\right]} - 2\sqrt{gh_0} \,. \tag{5}$$

Following the methodology developed in Didenkulova (2008), we let this nonlinearly deformed wave described by Eq. (5) run-up on a plane beach, characterized by the water depth $h(x) = -x\tan\alpha$. This approach does not take into account the merging point of the two bathymetries and, therefore, does not account for reflection from the toe of the slope and wave interaction with the reflected wave.

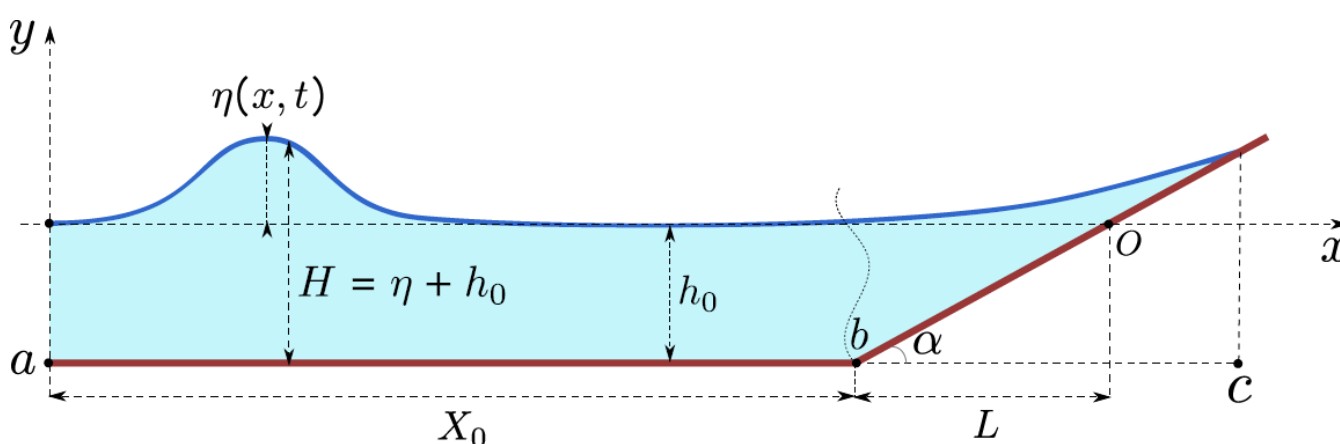

**Figure 1: Bathymetry sketch. The wavy curve at the toe of the slope regards analytical solution, which does not take into account merging between the constant depth and sloping beach sections.**

To do this we represent the input wave $\eta_{X0}$ as a Fourier integral:

$$\eta_{X0} = \int_{-\infty}^{+\infty} B(\omega)\exp(i\omega t)\,d\omega .$$
(6)

Its complex spectrum $B(\omega)$ can be found in an explicit form in terms of the inverse Fourier transform:

$$B(\omega) = \frac{1}{2\pi}\int_{-\infty}^{+\infty} \eta_{X0}(t)\exp(-i\omega t)\,dt .$$
(7)

Eq. (7) can be re-written in terms of the water displacement, produced by the wave maker at the left boundary (Zahibo et al.
2008):

$$B(\omega) = \frac{1}{2\pi i\omega}\int_{-\infty}^{+\infty}\frac{d\eta_0}{dz}\exp\left(-i\omega\left[z + \frac{x + X_0 + L}{V(\eta_0)}\right]\right)dz , \qquad z = t - \frac{x + X_0 + L}{V(\eta_0)} .$$
(8)

In this study we consider long single pulses of positive polarity:

$$\eta_0(t) = A\,\text{sech}^2\left(\frac{t}{T}\right),$$
(9)

where $A$ is the input wave height and $T$ is the effective wave period at the location with the water depth $h_0$. The wave described by Eq. (9) has an arbitrary height and period and, therefore, does not satisfy properties of the soliton, but just has a sech$^2$ shape. Substituting Eq. (9) into Eq. (8), we can calculate the complex spectrum $B(\omega)$.

Wave run-up oscillations at the coast $r(t)$ and the velocity of the moving shoreline $u(t)$ can be found from (Didenkulova et al. 2008):

$$r(t) = R\left(t + \frac{u}{g\tan\alpha}\right) - \frac{u^2}{2g}, \tag{10}$$

$$u(t) = U\left(t + \frac{u(t)}{g\tan\alpha}\right), \tag{11}$$

$$R(t) = \sqrt{2\pi\tau(L)} \int_{-\infty}^{+\infty} \sqrt{|\omega|}\, H(\omega) \exp\left\{i\left[\omega(t - \tau(L)) + \frac{\pi}{4}\mathrm{sign}(\omega)\right]\right\} d\omega, \tag{12}$$

$$U(t) = \frac{1}{\tan\alpha}\frac{dR}{dt}, \tag{13}$$

where $\tau = 2L/\sqrt{gh_0}$ is a travel time to the coast.

This solution we also compare with the run-up of a single wave of positive polarity described by Eq. (9) (without nonlinear deformation). The maximum run-up height $R_{\max}$ of such wave (9) can be found from (Didenkulova et al. 2008; Sriram et al. 2016):

$$\frac{R_{\max}}{A} = 2.8312\sqrt{\cot\alpha}\left(\frac{1}{gh_0}\left(\frac{2h_0}{\sqrt{3}T}\right)^2\right)^{1/4} \tag{14}$$

If the initial wave is soliton, Eq. (14) coincides with the famous Synolakis formula (Synolakis, 1987).

## 3. Numerical model

Numerically, we solve the nonlinear shallow water equations Eqs. (1),(2) written in a conservative form for a total water depth. We include the effect of the varying bathymetry (in space) and neglect all friction effects. However, the resulting numerical model will take into account for some dissipation thanks to the numerical scheme dissipation, which is necessary for the stability of the scheme and should not influence much run-up characteristics. Namely, we employ the natural numerical method, which was developed especially for conservation laws - the finite volume schemes.

The numerical scheme is based on the second order in space UNO2 reconstruction, which is briefly described in (Dutykh et al. 2011b). In time we employ the third order Runge-Kutta scheme with locally adaptive time steps in order to satisfy the CFL stability condition. The numerical technique to simulate the wave run-up was described previously in (Dutykh et al.

2011a). The bathymetry source term is discretized using the hydrostatic reconstruction technique, which implies the well-balanced property of the numerical scheme (Gosse, 2013).

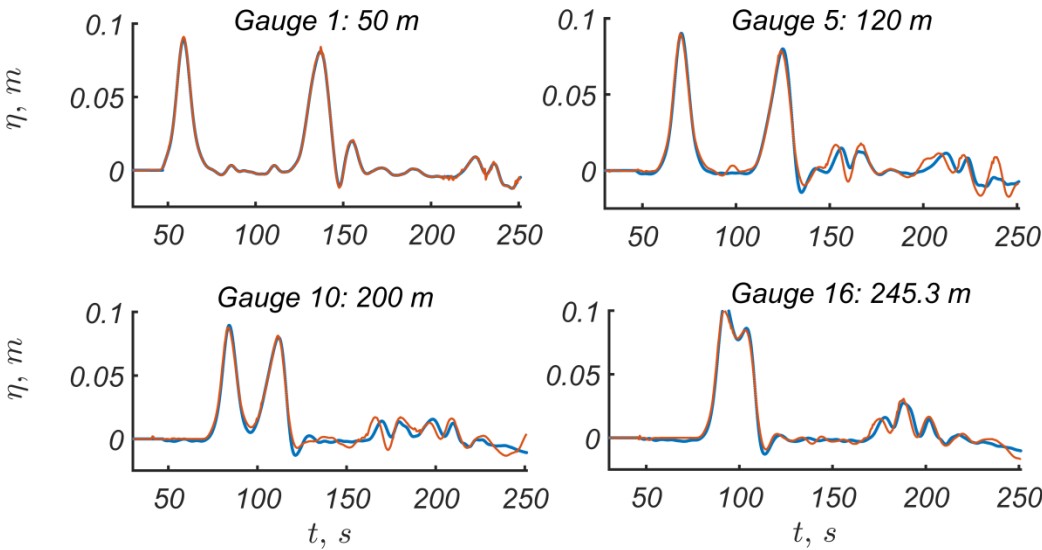

**Figure 2: Water elevations along the 251 m long constant depth section of the Large Wave Flume (GWK), $h_0 = 3.5$ m, $A = 0.1$ m, $T = 20$ s, $\tan\alpha = 1{:}6$: results of numerical simulations are shown by the red line, and experimental data are shown by the blue line.**

The numerical scheme is validated against experimental data of wave propagation and run-up in the Large Wave Flume

(GWK), Hannover, Germany. The experiments were set with a flat bottom with constant depth $h_0 = 3.5$ m and length $[a, b] = 251$ m, and a plane beach with a slope $\tan\alpha = 1{:}6$ (see Fig. 1). The flume had 16 wave gauges along the constant depth section and a run-up gauge on the slope. The incident wave had amplitude, $A = 0.1$ m, and period, $T = 20$ s. The detailed description of the experiments can be found in Didenkulova et al (2013). The results of numerical simulations are in a good agreement with the laboratory experiments as along the constant depth section (see Fig. 2) as also on the beach

(Fig. 3). The comparison of run-up height calculated numerically and analytically using the approach described in Section 2 with the experimental record in shown in Fig. 3. It can be seen that the experimentally recorded wave is slightly smaller which may be caused by the bottom friction and especially on the slope. Both numerical and analytical models describe the first wave of positive polarity rather well. The numerical prediction of run-up height is slightly higher than the analytical one. This additional increase in the run-up height in numerical model may be explained by the nonlinear interaction with the

reflected wave, which is not taken into account in the analytical model. The wave of negative polarity is much more sensitive to all the effects mentioned above than the wave of positive polarity and, therefore, looks different for all three lines in Fig. 3. By introducing additional dissipation in numerical model one can easily reach perfect agreement between the numerical simulations and experimental data. However, we do not do so, since below we are focusing on the analysis of analytical results and for clarity would like to avoid additional parameters in the numerical model. Also, we focus on the

maximum run-up height and, therefore, expect small differences between the results of analytical and numerical models.

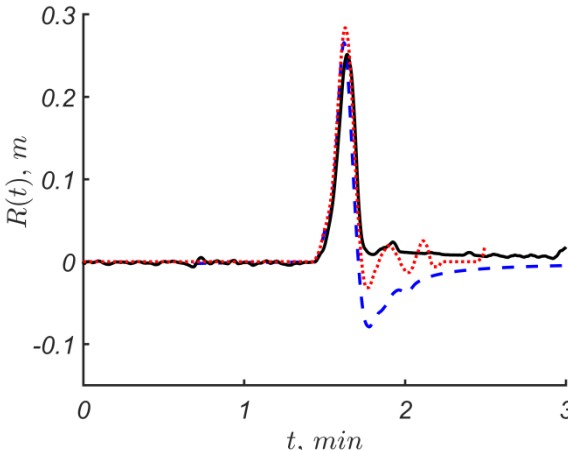

**Figure 3: Run-up height of the long single wave with $A = 0.1$ m and $T = 20$ s on a beach slope with tan $\alpha = 1:6$, the numerical solution is shown by the red dotted line, analytical solution is shown by the blue dashed line and the experimental record is shown by the black solid line.**

### 4. Results of numerical and analytical calculations

It is reported in (Didenkulova et al 2007; Didenkulova 2009) for a periodic sine wave, that the extreme run-up height increases proportionally to the square root of the wave front steepness. In this section, we study the nonlinear deformation and steepening of waves described by Eq. (9) and its effect on the extreme wave run-up height. The corresponding bathymetry used in analytical and numerical calculations is normalized on the water depth in the section of constant depth $h_0$, and is shown in Fig. 1. The input wave parameters such as wave amplitude, $A/h_0$, and effective wave length, $\lambda/X_0$, where $\lambda = T\sqrt{gh_0}$, are changed. The beach slope is taken tan $\alpha = 1:20$ for all simulations.

We underline that in order to have analytical solution, the criterion of no wave breaking should be satisfied. Therefore, all analytical and numerical calculations below are chosen for non-breaking waves.

Fig. 4 shows the dimensionless maximum run-up height, $R_{max}/A$ as a function of the initial wave amplitude, $A/h_0$. The incident wave propagates over different distances to the bottom slope, $X_0/\lambda = 1.7$, 3.4, 5.1, 6.8; $kh_0 = 0.38$. Analytical solution described in Section 2 is shown with lines, and numerical solution described in Section 3 is shown with symbols (diamonds, triangles, squares and circles). It can be seen that in most cases and especially for small values of $X_0/\lambda = 1.7$ and 3.4, numerical simulations give larger run-up heights than analytical predictions. These differences can be explained by the effects of wave interaction with the toe of the underwater beach slope, which are not taken into account in the analytical solution. For larger distances $X_0/\lambda = 6.8$, both analytical and numerical solutions give similar results, supported by the numerical scheme dissipation described in Section 3, which can be considered a "numerical error". It should be mentioned that we use zero physical dissipation rate for these simulations, however, small dissipation for stability of the numerical

scheme is still needed and this may become noticeable at large distances. For the sech²-shaped wave ($A/h_0 = 0.03$, $\lambda/X_0 = 0.12$) propagation, the reduction of initial wave amplitude constitutes ~2 %.

It is worth mentioning, that for small initial wave amplitudes all run-up heights are close to each other and are close to the thick black line, which corresponds to Eq. (14) for wave run-up on a beach without constant depth section. This means that the effects we are talking about are important only for nonlinear waves and irrelevant for weakly nonlinear or almost linear waves.

The same effects can be seen in Fig. 5, which shows the maximum run-up height, $R_{\max}/A$ as a function of distance to the

185 slope, $X_0/\lambda$, for different amplitudes of the initial wave, $A/h_0$. The distance $X_0/\lambda$ changes from 0.8 to 9.4; $kh_0 = 0.38$. The analytical solution is shown with lines while the numerical solution is shown with symbols (triangles, squares and circles). It can be seen in Fig. 5, for smaller values of $X_0/\lambda < 6$ numerical predictions provide relatively larger run-up values, as compared with analytical predictions, while for higher values of $X_0/\lambda > 6$ the differences are significantly reduced. A relevant change of this behaviour is given for $A/h_0 = 0.06$. We can observe that numerical predictions for this amplitude

become smaller than analytical predictions for $X_0/\lambda > 8$. As stated above, we believe that this can be a result of interplay of two effects: interaction with the underwater bottom slope, which is not taken into account in the analytical prediction and the numerical scheme dissipation ("numerical error"), which affects the numerical results.

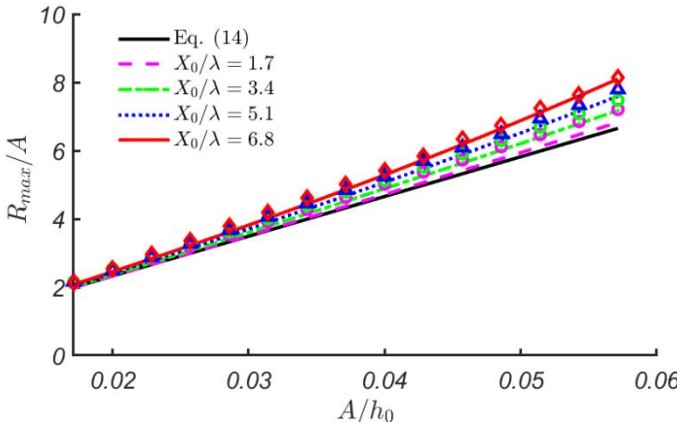

**Figure 4: Maximum run-up height, $R_{\max}/A$, as a function of initial wave amplitude, $A/h_0$, for different distances to the slope, $X_0/\lambda$. Analytical solution described in Section 2 is shown by lines and numerical solution described in Section 3 is shown by symbols (diamonds, triangles, squares and circles) with matching colours. The thick black line corresponds to Eq. (14) for wave run-up on a beach without constant depth section, $kh_0 = 0.38$.**

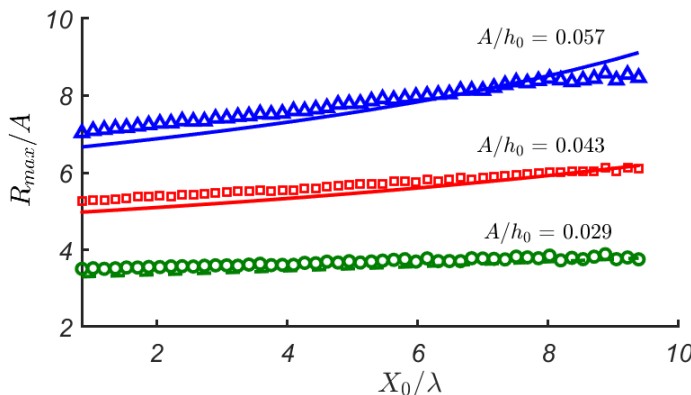

Figure 5: Maximum run-up height, $R_{max}/A$, as a function of distance to the slope, $X_0/\lambda$ for different amplitudes of the initial wave, $A/h_0$. Analytical solution described in Section 2 is shown by lines and numerical solution described in Section 3 is shown by symbols (triangles, squares and circles) with matching colours, $kh_0 = 0.38$.

The dependence of maximum run-up height, $R_{max}/A$ on $kh_0$ is shown in Fig. 6 for $A/h_0 = 0.03$. It can be seen that the difference between numerical and analytical results decreases with an increase in $kh_0$. We relate this effect with the wave interaction with the slope, which is not properly accounted in our analytical approach. As one can see in Fig. 7, this difference for a milder beach slope $\tan \alpha = 1:50$ is reduced.

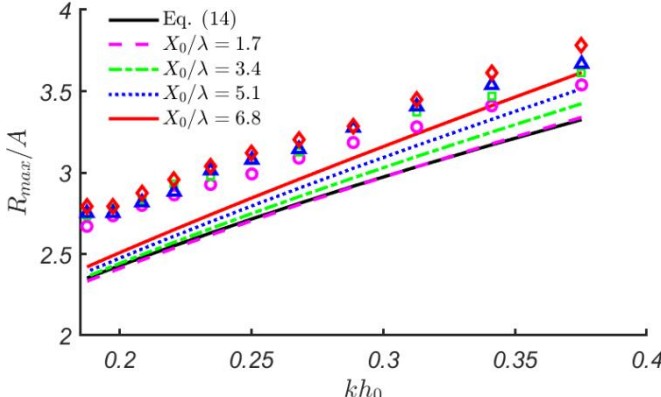

Figure 6: Maximum run-up height, $R_{max}/A$ as a function of $kh_0$ for different distances to the slope, $X_0/\lambda$. Analytical solution described in Section 2 is shown by lines and numerical solution described in Section 3 is shown by symbols (diamonds, triangles, squares and circles) with matching colours. The thick black line corresponds to Eq. (14) for wave run-up on a beach without constant depth section, $A/h_0 = 0.03$.

The next Fig. 8 supports all the conclusions drawn above. It also shows that difference between analytical and numerical results increases with an increase in wave period. As pointed out before for small wave periods the numerical solution may

coincide with the analytical one or even become smaller as it happens for $kh_0 = 0.38$ for $X_0 /\lambda > 8$.

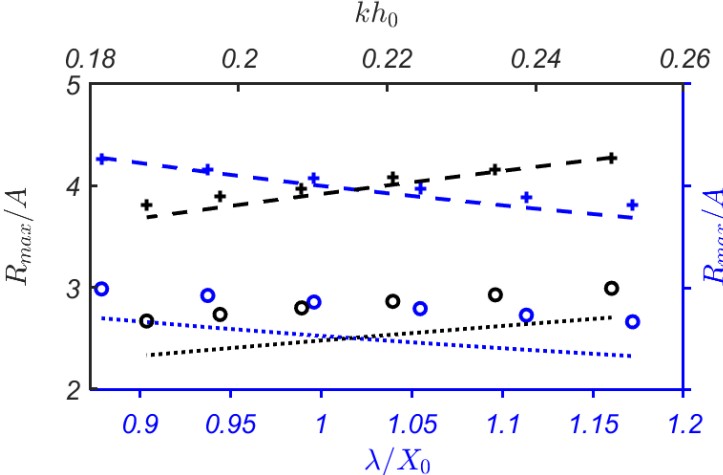

**Figure 7: Maximum run-up height, $R_{max}/A$ as a function of initial effective wave length, $\lambda/X_0$ (blue axes), and $kh_0$ (black axes). Analytical solutions for tan $\alpha = 1:20$ and tan $\alpha = 1:50$ are shown by dotted and dashed lines respectively, while numerical simulations for tan $\alpha = 1:20$ and tan $\alpha = 1:50$ are shown by circles and crosses respectively, $A/h_0 = 0.03$.**

Important, that both analytical and numerical results in Fig. 5 and Fig. 8 demonstrate an increase in maximum run-up height with an increase in the distance $X_0 /\lambda$. This result is in agreement with the conclusions of (Didenkulova et al 2007; Didenkulova, 2009) for sinusoidal waves. In order to be consistent with the results of (Didenkulova et al 2007; Didenkulova, 2009), we connect the distance $X_0 /\lambda$ with the incident wave front steepness in the beginning of the bottom slope. The wave front steepness $s$ is defined as maximum of the time derivative of water displacement, $d(\eta / A)/d(t/T)$, and is studied in relation with the initial wave front steepness, $s_0$, where:

$$s(x) = \frac{\max\left(d\eta(x,t)/dt\right)}{A/T}, \qquad s_0 = \frac{\max\left(d\eta(x=a,t)/dt\right)}{A/T}. \tag{15}$$

In order to calculate the incident wave front steepness in the beginning of the bottom slope from results of numerical simulations we should separate the incident wave and the wave reflected from the bottom slope. At the same time, the wave steepening along the basin of constant depth is very well described analytically as demonstrated in Fig. 9.

It can be seen that the wave transformation described by the analytical model is in a good agreement with numerical simulations. Therefore, below we reference to the analytically defined wave front steepness having in mind that it well coincides with the numerical one. Having said this, we approach the main result of this paper, which is shown in Fig. 10. The red solid line gives the analytical prediction. It is universal for single waves of positive polarity for different amplitudes $A/h_0$ and $kh_0$ and can be well approximated by the power fit (coefficient of determination $R$-squared = 0.99):

$$R_{max}/R_0 = \left(s/s_0\right)^{0.42}, \tag{16}$$

where $R_{max}/A$ is the maximum run-up height in the conjoined basin (with a section of constant depth), $R_0/A$ is the
corresponding maximum run-up height on a plane beach (without a section of constant depth).

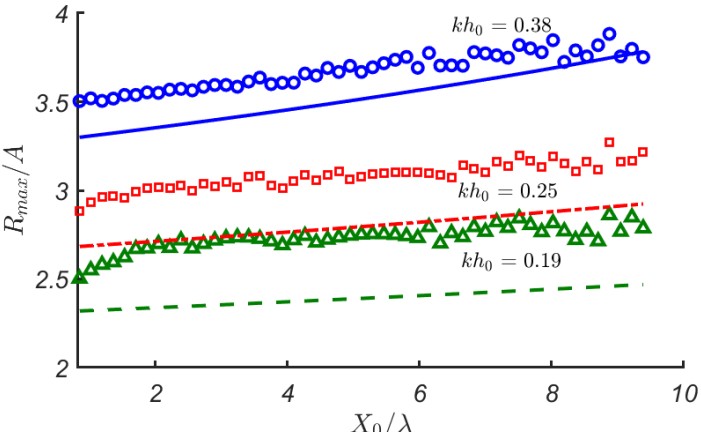

Figure 8: Maximum run-up height, $R_{max}/A$ as a function of the distance to the slope, $X_0/\lambda$ for different $kh_0$. Analytical solution described in Section 2 is shown by lines and numerical solution described in Section 3 is shown by symbols (triangles, squares and circles) with matching colours; $A/h_0 = 0.03$.

The fit is shown in Fig. 10 by the black dashed line. For comparison, the dependence of the maximum run-up height on the wave front steepness obtained using the same method for a sine wave is stronger than for a single wave of positive polarity (Didenkulova et al. 2007) and is proportional to the square root of the wave front steepness. This is logical as sinusoidal wave has a sign-variable form and, therefore, excites a higher run-up. For possible mechanisms, see discussion on $N$-waves
in (Tadepalli and Synolakis 1994).

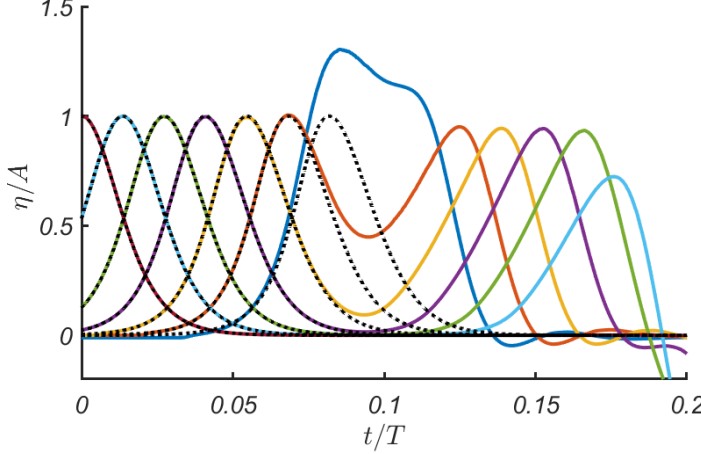

Figure 9: Wave evolution at different locations $x/\lambda = 0$, 0.85, 1.71, 2.56, 3.41, 4.27 and 5.12 along the section of constant depth for a basin with $X_0/\lambda = 5.12$ and $\tan \alpha = 1:20$. Numerical results are shown by solid lines, while the analytical predictions are given by the black dotted lines. The parameters of the wave: $A/h_0 = 0.03$, $kh_0 = 0.19$.

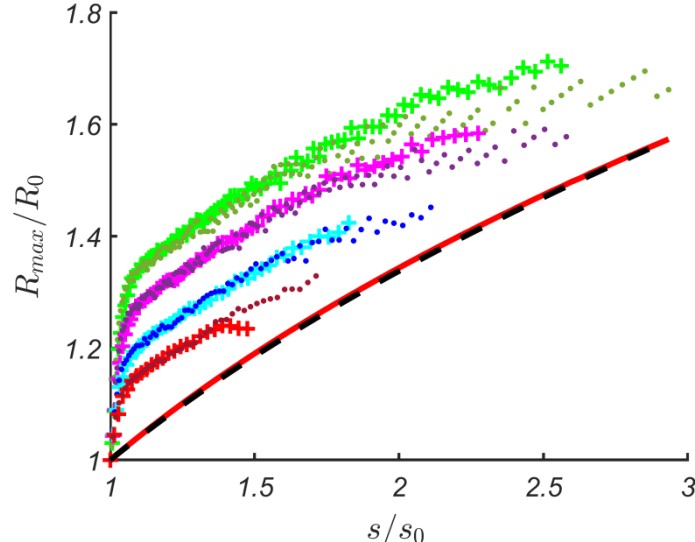

**Figure 10: The ratio of maximum run-up height in the conjoined basin, $R_{max}/A$ and the maximum run-up height on a plane beach, $R_0/A$ versus the wave front steepness, $s/s_0$ for $A/h_0 = 0.057$, $kh_0 = 0.38$ (brown points), $A/h_0 = 0.086$, $kh_0 = 0.38$ (red plus signs), $A/h_0 = 0.057$, $kh_0 = 0.29$ (blue points), $A/h_0 = 0.086$, $kh_0 = 0.29$ (turquoise plus signs), $A/h_0 = 0.057$, $kh_0 = 0.22$ (violet points), $A/h_0 = 0.086$, $kh_0 = 0.22$ (pink plus signs), $A/h_0 = 0.057$, $kh_0 = 0.19$ (dark green points), $A/h_0 = 0.086$, $kh_0 = 0.19$ (light green plus signs). All markers correspond to the results of numerical simulations, while the asymptotic analytical predictions are given by the red solid line. Black dashed line corresponds to the power fit of the analytical results Eq. (16).**

The results of numerical simulations are shown in Fig. 10 with different markers. It can be seen that numerical data for the same period, but different amplitudes follow the same curve. The run-up is higher for waves with smaller $kh_0$. In our opinion, this dependence on $kh_0$ is a result of merging plane beach with a flat bottom. This effect can be parameterized with the factor $(L/\lambda)^{1/4}$. The result of this parameterization is shown in Fig. 11. Here we can see that for smaller face front wave steepness, $s/s_0 < 1.5$, the run-up height is proportional to the analytically estimated curve shown by Eq. (16), while for larger face front wave steepness, $s/s_0 > 1.5$, the dependence on $s/s_0$ is weaker. This dependence for all numerical run-up height data, presented in Fig. 11, can be approximated by the power fit (coefficient of determination $R$-squared = 0.85):

$$R_{max}/R_0 = 1.17(\lambda/L)^{1/4}(s/s_0)^{1/4}. \tag{17}$$

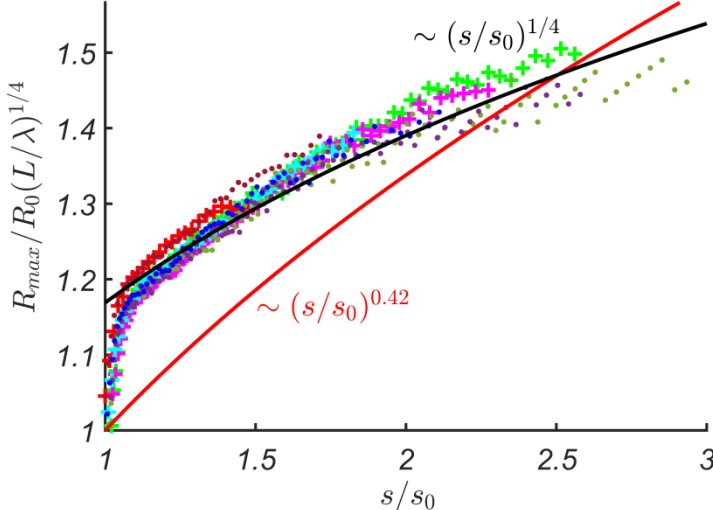

**Figure 11: The normalized maximum run-up height $R_{max}/R_0$ $(L/\lambda)^{1/4}$ calculated numerically versus the wave front steepness, $s/s_0$ for the same values of $A/h_0$ and $kh_0$ as in Figure 10. Red solid line is proportional to the "analytically estimated" Eq. (16), while black solid line corresponds to Eq. (17).**

## 5. Conclusions and Discussion

In this paper, we study the nonlinear deformation and run-up of tsunami waves, represented by single waves of positive polarity. We consider the conjoined water basin, which consists of a section of constant depth and a plane beach. While propagating in such basin, the wave shape changes forming a steep front. Tsunamis often approach the coast with a steep wave front, as it was observed during large tsunami events, e.g. 2004 Indian Ocean Tsunami and 2011 Tohoku tsunami.

The study is performed both analytically and numerically in the framework of the nonlinear shallow water theory. The analytical solution considers nonlinear wave steepening in the constant depth section and wave run-up on a plane beach independently and, therefore, does not take into account wave interaction with the toe of the bottom slope. The propagation along the bottom of constant depth is described by Riemann wave, while the wave run-up on a plane beach is calculated using rigorous analytical solutions of the nonlinear shallow water theory following the Carrier-Greenspan approach. The numerical scheme does not have this limitation. It employs the finite volume method and is based on the second order UNO2 reconstruction in space and the third order Runge-Kutta scheme with locally adaptive time steps. The model is validated against experimental data.

The main conclusions of the paper are the following.

- Found analytically, that maximum tsunami run-up height on a beach depends on the wave front steepness at the toe of the bottom slope. This dependence is general for single waves of different amplitudes and periods and can be approximated by the power fit: $R_{max} / R_0 = \left( s / s_0 \right)^{0.42}$.

- This dependence is slightly weaker than the corresponding dependence for a sine wave, proportional to the square root of the wave front steepness (Didenkulova et al. 2007). The stronger dependence of a sine wave run-up on the wave front steepness is consistent with the philosophy of *N*-waves (Tadepalli and Synolakis 1994).

- Numerical simulations in general support this analytical finding. For smaller face front wave steepness ($s/s_0 < 1.5$) numerical curves of maximum tsunami run-up height are parallel to the analytical one, while for larger face front wave steepness ($s/s_0 > 1.5$), this dependence is milder. The latter may be a result of numerical dissipation (error), which is larger for a longer wave propagation and, consequently, larger wave steepness. The suggested formula, which gives the best fit with the data of numerical simulations in general is $R_{\max}/R_0 = 1.17 \left(\lambda/L\right)^{1/4} \left(s/s_0\right)^{1/4}$.

- These results can also be used in tsunami forecast. Sometimes, in order to save time for tsunami forecast, especially for long distance wave propagation, the tsunami run-up height is not simulated directly, but estimated using analytical or empirical formulas (Glimsdal et al. 2019; Løvholt et al. 2012). In these cases we recommend using formulas, which take into account the face front wave steepness. The face front steepness of the approaching tsunami wave can be estimated from the data of the virtual (computed) or real tide-gauge stations and then be used to estimate tsunami maximum run-up height on a beach.

The nonlinear shallow water equations which are used in this study and commonly utilized for tsunami modelling, are also known as to neglect dispersive effects. In this context, it is important to mention the recent work of Larsen and Fuhrman (2019). They used RANS equations and k-ω model for turbulence closure to simulate propagation and run-up of positive single waves, including full resolution of dispersive short waves (and their breaking) that can develop near a positive tsunami front. They similarly showed that this effect depends on the propagation distance prior to the slope, if a simple toe with a slope type of bathymetry is utilized. This work shows that these short waves have little effect on the overall run-up, and hence give additional credence to the use of shallow water equations. These results largely confirm what was previously hypothesized by Madsen et al. (2008), that these short waves would have little effect on the overall run-up and inundation of tsunamis (though they found that they could significantly increase the maximum flow velocities).

**Acknowledgements**

The authors are very grateful to Professor Efim Pelinovsky, who gave an idea to this study a few years ago. Analytical calculations were performed with the support of RSF grant 16-17-00041. Numerical simulations and its comparison with the analytical findings were supported by the ETAG grant PUT1378. Authors also thank the PHC PARROT project No 37456YM, which funded the authors' visits to France and Estonia and allowed this collaboration.

**Data availability**

The data used for all figures of this paper are available at doi: 10.13140/rg.2.2.27658.41922. The source code (in Matlab)
used to generate this data may be shared upon request.

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
