# Peer review of "Nonlinear deformation and run-up of single tsunami waves of positive polarity: numerical simulations and analytical predictions"

_Natural Hazards and Earth System Sciences, 2019_

## Referee Comment (RC1) · Anonymous Referee #1 · 18 Jul 2019

The submitted manuscript considers the nonlinear deformation and run-up of single positive waves, both numerically and analytically. The work is based on solutions of nonlinear shallow water equations. Particular focus is on the effect of nonlinearity, and especially in the tsunami front steepness, on the eventual run-up. The work is generally clearly presented, but is not presented with sufficient generality to be widely interpreted (dimensional results are generally presented, which are then case specific, rather than a more proper dimensionless presentation which can be widely interpreted for tsunamis.) My suggestion is that the manuscript undergo a major revision, wherein the following specific comments are addressed:

Section 2: The paper considers solutions to nonlinear shallow water equations. These are commonly utilized to describe tsunamis, but are also known to neglect dispersive effects. In this context, the recent work of Larsen and Fuhrman (2019), https://doi.org/10.1016/j.coastaleng.2019.04.012, would be relevant to mention. They have similarly performed CFD simulations (RANS equations + k-omega model for turbulence closure) involving the run-up of positive single waves, including full resolution of dispersive short waves (and their breaking) that can develop near a positive tsunami front (there similarly shown that this effect depends on the propagation distance prior to the slope, if a simple toe with a slope type of bathymetry is utilized.) This work shows that these waves have little effect on the overall run-up, and hence give additional credence to the use of shallow water equations in the context of the present paper. These results largely confirmed what was previously hypothesized by Madsen et al. (2008), that these short waves would have little effect on the overall run-up and inundation of tsunamis (though they found that they could signficantly increase the maximum flow velocities).

line 103: A minor comment, but as the sech^2 variation is not periodic, variable T should probably be called an "effective period".

line 121: That numerical dissipation is included is introduced here, and later (e.g. lines 170-171 "supported by the numerical dissipation", and perhaps line 187, and elsewhere) this is seemingly taken as representing a positive and/or a physical effect. Numerical dissipation is model error (hence dependent on the space and time step, etc.), and should not be presented as otherwise. If dissipation effects are intended to be incorporated or considered, then an additional bottom friction term should be added.

lines 168-169: It is stated that the numerical simulations give larger run-up than predicted analytically, which is explained as due to the wave reflection from the underwater slope. As I understand it, this effect is neglected analytically, but included numerically. Would reflection from the slope then not reduce the incoming wave height and hence the run-up? Please clarify this apparent contradiction.

Nonlinear deformation. Starting with Fig. 4, results go essentially directly to the run-up, and seemingly skip the nonlinear deformation, though this aspect is emphasized in the title. Can the authors please present some snapshots of the nonlinear deformation leading to the run-up. This would give a much more comprehensive picture of what is being simulated.

Figs. 4, 5, 6, 7, 8, etc.: All of these figures are presented in dimensional terms, making their interpretation limited to the specific cases (wave characteristics and bottom slope) considered. This severely limits the overall potential impact of the paper. Please utilize a proper non-dimensional parameterization of the results. There are several potential ways to do this. As inspiration, it can be noted e.g. that Madsen and Fuhrman (2008) and Madsen and Schaffer (2010) have parameterized run-up R/H vs. surf similarity parameter \xi, and shown that the non-breaking run-up leads to a family of curves in terms of H/h at the toe. Some form of non-dimensional parameterization needs to be utilized, such that the results presented can be interpreted as widely as possible.

Similarly, Fig. 10 results seem only valid for the A and T combinations presented. These need to be presented in a way that can be more widely interpreted.

line 225: The wave front steepness s, is presented as the maximum of d\eta/dt i.e. as a velocity. Should the steepness not also be defined in a dimensionless manner?

---

## Referee Comment (RC2) · Anonymous Referee #2 · 23 Aug 2019

The topic of this paper is the analysis of the steepening of long single waves of positive polarity (LSWPP) and its effect on the wave run-up. It is studied from analytical and numerical approaches. After a review of the state of art in Section 1 (Introduction) the authors point out the novelty they present since it is the first time that the mentioned effect is addressed. Afterward, in section 2, they present the process they followed to obtain the analytical solution. Section 3 addresses the numerical modeling and comparison with the analytical solution. Section 4 presents the results of the analytical and numerical approaches. Finally, in section 5, the authors give a summary of the study and describe the conclusion they reached.

**GENERAL COMMENTS**

The topic is suitable for the journal since it addresses an issue of interest to the scientific community. The document is written clearly, it is up to the international standards and the length of the paper is adequate. The work is novel since it is the first time that the specific topic of the paper is addressed. However, some parts of the paper may be elaborated in more details and I have a few major concerns that recommend being addressed. These discussion points are outlined in the rest of this review:

1. The presented solution is valid for the applied wave (LSWPP). However, authors go further, linking this type of wave to a tsunami. In the introduction authors highlight that tsunamis have been represented in several ways, being some of them a soliton, like in the case they present. However, Madsen (2008) described how this simplification is not accurate enough to properly represent a tsunami, mainly because solitons do not take into account the tail of the tsunami that is the main component that determines the final run-up. Due to this, the use of tsunami word in the definition of the problem and the direct application of the results of the study to tsunamis must be directly addressed in the paper, explaining the limitations, especially after the paper of Madsen who broke the soliton paradigm for tsunamis. The presented solution could properly solve LSWPP, but its application to tsunamis need further discussion in the paper.

2. I understand from the manuscript that there is a limitation on the result because reflection is not included due to independent analysis: constant depth and plane beach. Does this limitation depend on the geometry (depth, amplitude, Xo) or is it a general limitation? Then in Figure 4 Xo is used and defined as the distance to the toe of the beach, so it is not clear if the real limitation of not taking reflection into account. In the case of a real tsunami, due to its wavelength, depending on the geometry, the reflection would certainly need to be included. Could you discuss this?

3. The presented analysis does not lack scientific rigor, but the generalization of the results is not direct. In order to make it easier the analysis of both graphs and results,
they must be non-dimensionalized. Some references of examples of this standardizations can be found in Madsen and Shaffer (2010), already referred by the authors. This modification allows a better analysis of the results and of the influence of each parameter in the final value of the run-up.

4. The final concern regards the application of the results to real tsunami cases, linked with my first comment. Have the authors tried to use/validate them in any real scenarios? I understand the limitation of the adopted LSWPP over an idealized geometry, but if the intention is to apply the results to tsunami scenarios, some details on the way to do this must be given.

5. Finally, section 5 of the paper is a summary of the work, together with a brief discussion. A summary is always interesting but a specific section of conclusions, clearly stated, for instance in bullets, would give a better overview.

---

## Author Comment (AC1) · 14 Oct 2019

**Response to the reviewer's comments on the manuscript "Nonlinear deformation and run-up of single tsunami waves of positive polarity: numerical simulations and analytical predictions" by Ahmed A. Abdalazeez et al.**

5  We thank the Reviewers #1 and #2 for their thorough work to improve our manuscript and useful comments and suggestions. We took into account all of them and revised the manuscript accordingly. The detailed point-by-point response is given below.

**Reviewer #1**

*Section 2: The paper considers solutions to nonlinear shallow water equations. These are commonly utilized to describe*
10  *tsunamis, but are also known to neglect dispersive effects. In this context, the recent work of Larsen and Fuhrman (2019), https://doi.org/10.1016/j.coastaleng.2019.04.012, would be relevant to mention. They have similarly performed CFD simulations (RANS equations + k-omega model for turbulence closure) involving the run-up of positive single waves, including full resolution of dispersive short waves (and their breaking) that can develop near a positive tsunami front (there similarly shown that this effect depends on the propagation distance prior to the slope, if a simple toe with a slope type of*
15  *bathymetry is utilized.) This work shows that these waves have little effect on the overall run-up, and hence give additional credence to the use of shallow water equations in the context of the present paper. These results largely confirmed what was previously hypothesized by Madsen et al. (2008), that these short waves would have little effect on the overall run-up and inundation of tsunamis (though they found that they could significantly increase the maximum flow velocities).*

The following paragraph was added in the end of the Conclusions and discussion Section:

20  "The nonlinear shallow water equations which are used in this study and commonly utilized for tsunami modelling, are also known as to neglect dispersive effects. In this context, it is important to mention the recent work of Larsen and Fuhrman (2019). They used RANS equations and k-ω model for turbulence closure to simulate propagation and run-up of positive single waves, including full resolution of dispersive short waves (and their breaking) that can develop near a positive tsunami front. They similarly showed that this effect depends on the propagation distance prior to the slope, if a simple toe
25  with a slope type of bathymetry is utilized. This work shows that these short waves have little effect on the overall run-up, and hence give additional credence to the use of shallow water equations. These results largely confirm what was previously hypothesized by Madsen et al. (2008), that these short waves would have little effect on the overall run-up and inundation of tsunamis (though they found that they could significantly increase the maximum flow velocities)."

*Line 103: A minor comment, but as the sech^2 variation is not periodic, variable T should probably be called an "effective*
30  *period".*

Corrected.

*Line 121: That numerical dissipation is included is introduced here, and later (e.g. lines 170-171 "supported by the numerical dissipation", and perhaps line 187, and elsewhere) this is seemingly taken as representing a positive and/or a*

*physical effect. Numerical dissipation is model error (hence dependent on the space and time step, etc.), and should not be*

35 *presented as otherwise. If dissipation effects are intended to be incorporated or considered, then an additional bottom friction term should be added.*

Corrected

*Lines 168-169: It is stated that the numerical simulations give larger run-up than predicted analytically, which is explained as due to the wave reflection from the underwater slope. As I understand it, this effect is neglected analytically, but included*

40 *numerically. Would reflection from the slope then not reduce the incoming wave height and hence the run-up? Please clarify this apparent contradiction.*

We corrected our statement. It is not only reflection but also interaction with the bottom slope and the reflected wave. By merging flat bottom with a plane beach we introduce the horizontal scale which in general case leads to resonant effects and may also increase the run-up height.

45 *Nonlinear deformation. Starting with Fig. 4, results go essentially directly to the runup, and seemingly skip the nonlinear deformation, though this aspect is emphasized in the title. Can the authors please present some snapshots of the nonlinear deformation leading to the run-up. This would give a much more comprehensive picture of what is being simulated.*

The propagation and wave deformation are shown in Fig. 9.

*Figs. 4, 5, 6, 7, 8, etc.: All of these figures are presented in dimensional terms, making their interpretation limited to the*
50 *specific cases (wave characteristics and bottom slope) considered. This severely limits the overall potential impact of the paper. Please utilize a proper non-dimensional parameterization of the results. There are several potential ways to do this. As inspiration, it can be noted e.g. that Madsen and Fuhrman (2008) and Madsen and Schaffer (2010) have parameterized run-up R/H vs. surf similarity parameter \xi, and shown that the non-breaking run-up leads to a family of curves in terms of H/h at the toe. Some form of non-dimensional parameterization needs to be utilized, such that the results presented can be*
55 *interpreted as widely as possible.*

We thank the referee for this comment. In the new version of the manuscript all parameters are dimensionless.

*Similarly, Fig. 10 results seem only valid for the A and T combinations presented. These need to be presented in a way that can be more widely interpreted.*

Done. Indeed, results in Fig. 10 do not depend on *A*, but on *T*. This was resolved by introducing the factor $(L/\lambda)^{1/4}$ in the new
60 version of the manuscript (see Fig. 11).

*Line 225: The wave front steepness s, is presented as the maximum of d\eta/dt i.e. as a velocity. Should the steepness not also be defined in a dimensionless manner?*

It has been normalized by *A*/*T*.

**Reviewer #2**

65 *1.        The presented solution is valid for the applied wave (LSWPP). However, authors go further, linking this type of wave to a tsunami. In the introduction authors highlight that tsunamis have been represented in several ways, being some of them a soliton, like in the case they present. However, Madsen (2008) described how this simplification is not accurate enough to properly represent a tsunami, mainly because solitons do not take into account the tail of the tsunami that is the main component that determines the final run-up. Due to this, the use of tsunami word in the definition of the problem and*

70    *the direct application of the results of the study to tsunamis must be directly addressed in the paper, explaining the*
*limitations, especially after the paper of Madsen who broke the soliton paradigm for tsunamis. The presented solution could*
*properly solve LSWPP, but its application to tsunamis need further discussion in the paper.*

We acknowledge the "soliton paradigm" described by Madsen et al. (2008) as not applicability of solitons (mathematical solutions of Korteweg de Vries equation) to the tsunami problem due to their short wave length compare to typical tsunamis.
75    Regarding the tsunami wave shape, it varies very much and cannot be unified.

We added the following text to the Introduction on P.2:

"It should be mentioned that the shape of tsunami varies a lot depending on its origin and the propagation path. One of the best examples of tsunami wave shape variability is given in Shuto (1985) for the 1983 Japan Sea tsunami, where the same tsunami event resulted in very different tsunami approaches in different locations along Japanese coast. These wave shapes
80    included: single positive pulses, undergoing both surging and spilling breaking scenarios, breaking bores, periodic wave trains, surging and breaking as well, a sequence of two or three waves and undular bores. This is why there is no such term as "typical tsunami wave shape", and therefore in the papers on wave run-up cited above many different wave shapes, such as single pulses, *N*-waves, periodic symmetric and asymmetric wave trains, are considered."

2.       *I understand from the manuscript that there is a limitation on the result because reflection is not included due to*
85    *independent analysis: constant depth and plane beach. Does this limitation depend on the geometry (depth, amplitude, Xo)*
*or is it a general limitation? Then in Figure 4 Xo is used and defined as the distance to the toe of the beach, so it is not clear*
*if the real limitation of not taking reflection into account. In the case of a real tsunami, due to its wavelength, depending on*
*the geometry, the reflection would certainly need to be included. Could you discuss this?*

The limitation exists only in the analytical approximation. The study also includes direct numerical simulations using
90    nonlinear shallow water equations, which does not have such limitation.

3.       *The presented analysis does not lack scientific rigor, but the generalization of the results is not direct. In order to*
*make it easier the analysis of both graphs and results, they must be non-dimensionalized. Some references of examples of*
*this standardizations can be found in Madsen and Shaffer (2010), already referred by the authors. This modification allows*
*a better analysis of the results and of the influence of each parameter in the final value of the run-up.*

95    We thank the referee for this comment. In the new version of the manuscript all parameters are dimensionless.

4.       *The final concern regards the application of the results to real tsunami cases, linked comment with my first*
*comment. Have the authors tried to use/validate them in any real scenarios? I understand the limitation of the adopted*
*LSWPP over an idealized geometry, but if the intention is to apply the results to tsunami scenarios, some details on the way*
*to do this must be given.*

100   We added the following comment in the Conclusions and Discussion:

"These results can also be used in tsunami forecast. Sometimes, in order to save time for tsunami forecast, especially for long distance wave propagation, the tsunami run-up height is not simulated directly, but estimated using analytical or empirical formulas (Glimsdal et al. 2019; Løvholt et al. 2012). In these cases we recommend using formulas, which take into

account the face front wave steepness. The face front steepness of the approaching tsunami wave can be estimated from the data of the virtual (computed) or real tide-gauge stations and then be used to estimate tsunami maximum run-up height on a beach."

5.     *Finally, section 5 of the paper is a summary of the work, together with a brief discussion. A summary is always interesting but a specific section of conclusions, clearly stated, for instance in bullets, would give a better overview.*

Done

[revised manuscript text omitted]